# RET Proto-Oncogene—Not Such an Obvious Starting Point in Cancer Therapy

**DOI:** 10.3390/cancers14215298

**Published:** 2022-10-27

**Authors:** Tomasz Kucharczyk, Paweł Krawczyk, Dariusz M. Kowalski, Adam Płużański, Tomasz Kubiatowski, Ewa Kalinka

**Affiliations:** 1Chair and Department of Pneumonology, Oncology and Allergology, Medical University of Lublin, 20-059 Lublin, Poland; 2Department of Lung and Thoracic Tumours, Maria Skłodowskiej-Curie National Research Institute, 02-718 Warsaw, Poland; 3Oncology and Immunology Clinic, Warmian-Masurian Cancer Center of the Ministry of the Interior and Administration’s Hospital, 10-228 Olsztyn, Poland; 4Department of Oncology, Polish Mother’s Memorial Hospital-Research Institute, 90-302 Lodz, Poland

**Keywords:** RET, thyroid cancer, non-small cell lung cancer, selpercatinib, pralsetinib

## Abstract

**Simple Summary:**

Rearranged during transfection (RET) is a transmembrane receptor tyrosine kinase that shows targetable mutations and gene fusions in a few types of cancers. Multiple kinase inhibitors, which target RET kinase, show little efficiency in tumors with *RET* gene alterations. Therefore, the new RET-specific inhibitors have been developed. Selpercatinib and pralsetinib showed high effectiveness in clinical trials, but their efficacy was shown to be reduced by the occurrence of secondary alterations such as resistance mutations or activation of different signaling pathways. New drugs that could be used in patients with secondary mutated *RET* gene are under investigation and show promising results.

**Abstract:**

Mutations and fusions of *RET* (rearranged during transfection) gene are detected in a few common types of tumors including thyroid or non-small cells lung cancers. Multiple kinase inhibitors (MKIs) do not show spectacular effectiveness in patients with *RET*-altered tumors. Hence, recently, two novel *RET*-specific inhibitors were registered in the US and in Europe. Selpercatinib and pralsetinib showed high efficacy in clinical trials, with fewer adverse effects, in comparison to previously used MKIs. However, the effectiveness of these new drugs may be reduced by the emergence of resistance mutations in *RET* gene and activation of different activating signaling pathways. This review presents the function of the normal RET receptor, types of molecular disturbances of the *RET* gene in patients with various cancers, methods of detecting these abnormalities, and the effectiveness of modern anticancer therapies (ranging from immunotherapies, through MKIs, to RET-specific inhibitors).

## 1. Introduction

Molecular targets in cancer treatment have been known for quite a long time already and the list of drugs used with specific genetic changes is growing each year. The most common targets for such therapies are between signal cascades, whether these are receptors in the cancer cell membrane or some downstream signal transmitting molecules. The binding sites for modern drugs are receptor tyrosine kinases (RTK) or serine-threonine kinases of intracellular transmitters. These targets are altered by oncogenic driver abnormalities. In many types of tumors, we are able to detect the same molecular changes. Hence, it is possible to use the same molecularly targeted therapy for different oncologic disorders (tissue agnostic therapies). Such molecular changes appear in a small percentage of cancer patients, but with novel drugs and a wider target spectrum, it is possible to detect such a change in a bigger group of those that need the treatment.

The most commonly targeted molecules are tyrosine kinases which are responsible for the activation of many signaling pathways, resulting in neoplastic transformation. The *HER2* (human epidermal growth factor receptor 2), EGFR (epidermal growth factor receptor) and KIT (KIT proto-oncogene) were the first target for tyrosine kinases inhibitors (TKIs) discovered more than 15 years ago. Since the development of the first generation of these therapies, this group of drugs has been extended to include therapies targeting ALK (anaplastic lymphoma kinase), VEGFR (vascular endothelial growth factor receptor), ROS1 (ROS proto-oncogene 1), MET (mesenchymal-epithelial transition factor), PDGFR (platelet derived growth factor receptor), NTRK 1-3 (neurotrophic tyrosine kinase type 1-3), or RET (RET proto-oncogene, rearranged during transfection) [1,2].

RET is a transmembrane receptor tyrosine kinase, which is coded by proto-oncogene *RET*, and was first discovered in T cell lymphoma [3]. In normal cells, RET is involved in the process of fetal development of hematopoietic, genitourinary, gastrointestinal, and nervous systems. However aberrations of *RET* are known to be oncogenic drivers in many types of cancers. There are two models of RET oncogenic activation, namely gene fusions and point mutation. While gene fusions are more common in papillary thyroid carcinoma (PTC), in around 10–20% of patients, they are much less common in non-small cell lung cancer (NSCLC), in around 1–2% of patients. On the other hand, point mutations in *RET* gene are common in sporadic medullary thyroid cancer (MTC), in 60–90% of patients, and in multiple endocrine neoplasia syndrome type 2 (MEN2) [3,4,5].

Aberrant RET has been the target of few previously developed multiple kinase inhibitors such as alectinib, vandetanib, regorafenib, or cabozantinib [2]. Since these drugs are not specifically RET-directed, they did not show spectacular results in RET-aberrant patients, and the process of development of RET-directed drugs is ongoing. Currently, there are two selective RET TKIs registered by the US Food and Drug Administration (FDA) and European Medicines Agency (EMA) and a few more in early clinical trials. In this summary, we present an overview of *RET* gene changes, their implications, and the latest reports on treatment options for RET-aberrant patients.

## 2. RET Protein and Activity

The *RET* proto-oncogene is located on chromosome 10 (10q11.21) and consists of 21 exons [6]. The protein encoded by the gene has three distinct elements: an extracellular domain, with characteristic cadherin-like domains, a transmembrane domain, and an intracellular domain, with tyrosine kinase activity. The extracellular fragment of the protein has four domains that are structurally similar to cadherin molecules, hence the name cadherin-like domain (CLD 1-4). Between CLD2 and CLD3, there is a Ca^2+^ binding site, which is necessary for the proper functioning of RET. Below the CLDs, a cysteine-rich fragment is located, spanning 120 amino acids. There are three isoforms of RET protein resulting from alternative splicing, consisting of 1072, 1106, and 1114 amino acids, which have short (RET9), intermediate (RET43), and long (RET51) amino acid carboxyl-terminal tails, respectively. The short and long forms are known to have higher expression than the intermediate form, and the RET51 form is considered to be most active in tumorigenesis and is believed to be more common in MEN2 tumors [7,8,9,10]. Activation of RET may be initiated by different ligands: growth/differentiation factor 15 (GDF15), glial cell line-derived neurotrophic factor (GDNF), GDNF receptor α (GFRα)-like protein (GFRAL), artemin, neurturin, persephin, and four coreceptors, namely GDNF family receptor-α (GFRα1-4). Following the activation, phosphorylation of different tyrosine residues results in activation of different signaling pathways downstream of RET. Previous studies have shown that 14 out of 18 Tyr residues may undergo phosphorylation, of which Tyr^900^ and Tyr^905^ occurring in the kinase activation loop, are known to contribute to full kinase activation. Autophosphorylation of residue Tyr^1062^ is crucial for activation of RAS/MAPK and PI3K/AKT pathways. The Tyr^1096^ residue phosphorylation on RET51 isoform contributes also to the activation of these pathways, but it does not in the RET9 isoform [11]. Phosphorylation of Tyr^1015^ mediates the binding of phospholipase C, and in effect activation of protein kinase C. These pathways play an important role in the differentiation, proliferation, cell migration, and survival of the cell [12].

*RET* gene abnormalities were shown to be one of the driving alterations in quite a few diseases, including cancers. Activating point mutations have been described in a hereditary cancer syndrome multiple endocrine neoplasia type 2, in which medullary thyroid carcinoma and pheochromocytoma may develop, or in familial medullary thyroid cancer (FMTC). On the other hand, *RET* rearrangements are described in papillary thyroid carcinoma, non-small cell lung carcinoma, and other sporadic types of cancers. Gain of function aberrations are present mostly in oncogenic diseases, but RET may also be inactivated, causing Hirschsprung’s disease, congenital kidney or urinary tract anomalies, or innate central hypoventilation syndrome [13,14].

## 3. RET Rearrangements

Somatic *RET* gene rearrangements usually involve a 3′ sequence of the *RET* gene, which codes the kinase domain, and the 5′ sequence of other partner genes. The most common breakpoint localization is within intron 11 and it leads to fusions with only the cytoplasmic fragment of the protein. Other breakpoints may occur in introns 7 and 10, where the rearranged form involves the transmembrane domain [15]. There are already more than 35 partner fusion genes described [16]. Oncogene activation can take place by two mechanisms: either the fusion partner shares its dimerization domain, which connects with the kinase domain of *RET*, which results in ligand-independent constitutive activation, or the kinase domain of *RET* is controlled by the promoter region of the fusion gene [4,17]. Similarly to other driving alterations in cancers, *RET* rearrangements usually occur exclusively with other activating changes, such as *EGFR* gene mutations or *ALK* and *ROS1* gene rearrangements, although they might appear as a result of treatment with different tyrosine kinase inhibitors in patients with other sensitizing molecular abnormalities [18,19]. The most common mutations and gene fusion partners are presented in Figure 1.

*RET* gene rearrangements are most common among PTC, with a prevalence between 5% and 35% of patients, although patients who were previously exposed to radiation have these aberrations detected in 50–80% of cases. The most recent analysis of *RET* fusions in PTC by The Cancer Genome Atlas consortium (TCGA) found that the prevalence of those alterations is detected in approximately 6.8% of cases [20,21,22,23,24,25]. Fusions in PTC are described as RET/PTC, signed with consecutive numbers. The most common fusion genes involving the *RET* gene are as follows: the coiled-coil domain containing 6 *CCDC6-RET (RET/PTC1)* and nuclear receptor coactivator 4 *NCOA4-RET (RET/PTC3)*, the latter being believed to cause more severe disease course [26]. Most partner genes observed in PTC are as follows: *AKAP13*, *ERC1*, *FKBP15*, *GOLGA5*, *HOOK3*, *KTN1*, *PCM1*, *PRKAR1A*, *RFG9*, *SPECC1L*, *TB1LXR1*, *TRIM24*, *TRIM27*, *TRIM33*. These genes form interchromosomal translocations. *CCDC6* and *NCOA4,* however, are located on chromosome 10 as well, forming paracentric inversions [27]. Most of these fusion proteins are detected in the cytosol, hence they avoid regular transportation by endosomes and ubiquitin-mediated degradation in lysosomes [28].

*RET* fusion can be also detected in approximately 2% of NSCLC cases, mainly in young (≤60 y.o.), minimal or non-smoking, adenocarcinoma patients [29,30]. The most frequently observed rearrangements are with *KIF5B* gene, forming an intrachromosomal pericentric inversion, and with *CCDC6*. Other partner genes may be *NCOA4*, *EPHA5*, *MYO5C*, *TRIM33*, *CLIP*, *ERC1*, *PICALM*, *FRMD4A*, and *RUFY2*. *KIF5B-RET* fusion is believed to increase RET expression by even 30 times [26,31,32].

Papillary thyroid carcinoma and non-small cell lung cancer are not the only neoplasms that present *RET* rearrangements. These aberrations are also detected in breast cancer, colon cancer, salivary intraductal carcinomas, chronic myelomonocytic leukemia, or spitz tumors [5,33,34,35,36,37].

## 4. Mutations

Germline mutations of *RET* gene are responsible for the cases of autosomal dominant hereditary multiple endocrine neoplasia type 2 (MEN2A and MEN2B), as well as familial medullary thyroid carcinoma. These syndromes are associated with the development of medullary thyroid carcinoma (almost 100% of cases), pheochromocytoma (around 50% of cases), or primary hyperparathyroidism, in family members who are affected. MEN2B is associated with a more severe course of the disease, with MTC developing at an early age, and being more aggressive. On the other hand, patients with FMTC present the mildest forms of MTC, with very rare or without other endocrinopathies [38,39].

*RET* mutations can be divided into two groups: the ones present in the extracellular, cysteine-rich domain (CRD), and the ones present in the tyrosine kinase domain. The mutations occurring in the CRD fragment are mostly located in exons 8, 10 (codons 610, 611, 618, or 620), and 11 (codons 630 or 634) [40,41,42], and result in dimerization of mutant RET monomers and consecutive ligand-independent activation of the receptor [43,44]. Mutations in the cysteine-rich domain are more common for MEN2A and FMTC, with C634R in exon 11 representing 85% of detected mutations, whereas the tyrosine kinase domain (TKD) mutations are present more frequently in MEN2B. Mutations in exons 13–16 alter the activity of the tyrosine kinase domain, with the most common M918T mutation (95% of cases) in exon 16 and A883F (2–3% of cases) in exon 15 being the most aggressive. TKD mutations result in increased ATP binding and activation of receptors regardless of monomer dimerization [45,46,47]. Changes in intracellular fragments of *RET* gene are also detected in MEN2A and FMTC (most commonly exon 13 E768D and L790F mutations, and exon 14 V804L and V804M mutations), and are associated with later disease development.

Somatic *RET* gene activating mutations are present also in a variety of cancer, including desmoplastic melanoma (20%), cutaneous squamous cell carcinoma (10%), melanoma (6.6%), colorectal carcinoma (3.6%–6.9%), breast cancer, urothelial carcinoma, or paraganglioma [5,16,48].

Loss of function mutations of *RET* gene can be detected in diseases of non-malignant character. These are most commonly described in Hirschsprung’s disease (HSCR), a congenital malformation with the presence of aganglionosis of the gastrointestinal tract. *RET* mutations occur in approximately 50% of familial and 10–20% of sporadic cases of HSCR [14,49,50,51]. There are different mechanisms of RET impairment and consequently loss of its function, mainly depending on the localization of mutation. The CLD and CRD fragments mutations result in misfolding of the protein and lack of surface expression, whereas mutations in TKD result in loss of kinase activity [52,53].

## 5. Detection of RET Alterations

Genetic changes in *RET* gene are easily detected using relatively cheap and simple molecular methods, mainly real-time polymerase chain reaction (PCR), immunohistochemistry (IHC), and fluorescent in-situ hybridization (FISH). However, these methods do not allow the detection of all fusion variants and rare mutations in the RET gene. With the development of large-scale sequencing methods, next-generation sequencing (NGS) platforms allow for the detection of known, but also new mutations, amplifications, and fusions/rearrangements in one reaction. Even though the NGS method is the most sensitive and the most specific out of the possibilities, it is also the most expensive. FISH and IHC have varied results in the detection of RET fusions. It is also worth noticing that only NGS analysis allows for detection of many aberrations in a single reaction and regarding the broad spectrum of these changes, it is considered to be the best choice in their detection. The most commonly used samples are tumor biopsies, but liquid biopsy (peripheral blood) is an encroaching method, allowing for quick collection of circulating free DNA and its consecutive analysis for predictive factors [29,31,54,55,56,57,58].

## 6. Immunotherapy in Tumors with Genetic Changes of RET Gene

Studies on *RET*-altered NSCLC cases showed little to no effect of immunotherapeutics. The study of Hegde et al. showed that patients with *RET* fusions or mutations did not achieve any significant effects from immunotherapy with immune checkpoints inhibitors (ICIs) when compared to other treatment methods. Time to disease discontinuation was only 5.2 months in the group that received ICIs and 18 months in the non-ICIs group [59]. Results of the study carried out at Memorial Sloan Kettering Cancer Center showed that lung cancer patients with *RET* alterations showed no response to immune checkpoint inhibitors with a median progression-free survival (mPFS) of 3.4 months [60]. In the IMMUNOTARGET study, in which NSCLC patients with oncogenic driving alterations were treated with ICIs in the second and subsequent lines, patients with *RET*-altered NSCLC achieved median overall survival (mOS) 21.3 months (3.8; 28.0 mo) and mPFS 2.1 months (1.3; 4.7 mo). Among 16 patients with *RET* rearrangement, 1 patient (6.25%) responded to ICIs therapy, and short-term disease stabilization occurred in 3 patients (18.75%). 75% of patients receiving ICIs showed disease progression [61]. Retrospective analysis of 13 Korean patients with *RET* fusions showed mPFS of 2.1 months and mOS of 12.4 months. The overall response rate reached only 7.7% [62]. Lung tumors with *RET* fusions usually appear in patients without a smoking history. Such tumors tend to have a low tumor mutation burden (TMB) and are poorly infiltrated by immunological cells, which in effect present low expression of PD-L1, and hence weak response to ICIs [63].

## 7. Treatment of RET-Altered Cancers with Multiple Kinase Inhibitors

Due to the fact, that RET has a tyrosine kinase activity, it might be the target of multiple kinase inhibitors (MKIs). Several of these molecules have been studied, but in most cases, the specific interaction with *RET*-altered tumors has not been described [1,13]. Studies with MKIs in *RET*-altered NSCLC have shown varied results, with an overall response rate (ORR) between 16 and 47%, mPFS between 4.9 and 7.3 months, and mOS between 4.9 and 11.6 months. However, it is worth noticing that between 53% and 73% of the patients in these studies had a dose reduction of MKIs, and 8–21% patients discontinued the treatment, due to at least grade 3 adverse effects [64,65,66,67].

MKIs approved by the FDA and EMA for the treatment of radioiodine refractory differentiated thyroid cancer (DTC) are sorafenib and lenvatinib [68,69]. However, lenvatinib has been proven to be more effective than sorafenib (mPFS reached 35.3 vs. 13.3 months, respectively) [70]. Vandetanib and cabozantinib are approved for the treatment of metastatic MTC based on ZETA and EXAM clinical trials. The ZETA trial showed that mPFS in vandetanib arm was 30.5 vs. 19.3 months in the placebo arm, with an ORR of 45% vs. 13%, respectively. Patients with M918T mutation presented higher ORR than those without this mutation (54.5% vs. 30.9%, respectively). The EXAM trial showed 11.2 months of mPFS in cabozantinib-treated patients vs. 4.0 months in patients receiving placebo and ORR of 28% vs. 0%, respectively. The post-trial analysis also showed that cabozantinib had higher efficacy in patients with M918T mutation than in those not presenting such *RET* gene alteration (mPFS was 14.2 vs. 5.8 months, mOS was 44.3 vs. 20.2 months and ORR was 34% vs. 2%, respectively) [71,72,73].

The effectiveness of MKIs in *RET*-altered cancers, although modest in NSCLC patients and slightly higher in MTC patients, is overshadowed by commonly present adverse effects and toxicities, which result in dose reduction or even treatment discontinuation. This may arise from off-target inhibition of VEGFR1 and VEGFR2, or EGFR [13]. The most recently developed MKIs, with anti-RET activity is RXDX-105, which inhibits both RET and BRAF, and spares VEGFR1/2, resulting in fewer side-effects than previously developed MKIs. Its effectiveness was studied in a Phase 1/1b trial and presented activity against various forms of *RET* fusion and mutations in cancer patients. Interestingly, RXDX-105 did not show efficiency in NSCLC patients with *KIF5B-RET* fusion and V804L/M gatekeeper mutations, although when administered in non-*KIF5B-RET* fusion patients, the ORR was 67% [46,74,75].

## 8. RET-Specific Inhibitors

In recent years, two novel inhibitors targeting RET-altered tumors, have been developed. Selpercatinib was approved by the FDA and the EMA in treatment of adult patients with advanced NSCLC, and patients of ≥12 years of age with advanced or metastatic thyroid carcinoma. Pralsetinib was approved by the FDA in the treatment of advanced NSCLC and thyroid cancer, but the EMA so far approved this drug only for NSCLC patients [76,77,78,79,80].

Pralsetinib (BLU-667) is a small-molecule selective inhibitor of RET, with a mechanism of action based on blocking the kinase domain and bypassing the gatekeeper mutations (V804L/M), which for example block the action of vandetanib, as well as presenting effectiveness in patients with *RET* fusions. The drug presents high selectivity against VEGFR2, decreasing off-target adverse effects [81]. The phase 1/2 registrational ARROW trial showed good tolerability, with only 4% of treatment discontinuations due to treatment-related side effects. NSCLC patients with *RET* fusions, treated with pralsetinib, showed 73% ORR in the treatment-naïve group and 61% ORR in the group with previously administered chemotherapy. Pralsetinib also presented effectiveness in *RET* fusion-positive brain metastases, where 78% of patients with intracranial metastases showed shrinkage of the metastatic tumor, proving penetration of the molecule to the central nervous system (CNS) [82]. The update to the study showed that the treatment-naïve patients reached 79% ORR, with mPFS of 13 months, while the chemotherapy patients previously treated with platinum achieved 62% ORR and mPFS of 16.5 months [83].

The study also analyzed patients with *RET*-mutated and *RET* fusion-positive MTC patients, where ORR for previously treated, treatment-naïve and fusion-positive previously treated thyroid cancer patients were 60%, 71%, and 89%, respectively [84,85]. The ARROW study also showed that pralsetinib was active in other types of *RET*-positive solid tumors, including other thyroid tumors, cholangiocarcinoma, pancreatic and neuroendocrine tumors. 11 patients with thyroid tumors had 91% ORR and the disease control rate (DCR) was 100%. Patients with other tumor types had ORR of 50% and DCR of 92%. It was also presented that circulating tumor DNA (ctDNA) was effectively removed from plasma during the treatment with pralsetinib (≥50% of ctDNA in 90% of NSCLC patients and 83% in MTC patients).

The most common adverse events (AEs) in the whole studied group were as follows: increased aspartate aminotransferase (AST) (31%), anemia (22%), increased alanine aminotransferase (ALT) (21%), constipation (21%), hypertension (20%), and neutropenia (19%). Grade 3/4 hypertension and neutropenia were observed in 10% of cases, and anemia in 8% of studied cases [86,87]. In the NSCLC group, 93% of patients had any AEs, and the serious grade 3/4 AEs were observed in 48% of cases, among which the most common ones were: neutropenia (19%), hypertension (11%), and anemia (10%). Moreover, 38% of patients had their drug does reduced, and 6% had their treatment terminated due to AEs [82].

An international, randomized, open-label, phase 3 AcceleRET study, for patients with metastatic NSCLC, comparing pralsetinib with chemotherapy in non-squamous histology and pralsetinib with chemotherapy or immunotherapy in squamous histology, is currently underway (NCT04222972) [88].

Selpercatinib (LOXO-292) is the second recently registered small-molecule RET inhibitor. Similarly to pralsetinib, it is orally bioavailable and presents activity in gatekeeper mutated, *RET*-altered tumors [89]. The registrational, Phase 1/2 LIBRETTO-001 clinical trial showed the efficacy of the molecule in both MTC and NSCLC patients. In *RET*-mutated MTC patients, who previously were treated with MKIs, the ORR was 69%, with 9% of complete responses (CR). 1-year PFS was achieved in 82% of cases. In previously untreated MTC patients, the ORR was 73%, CR was observed in 11% of patients, mPFS was 23.6 months, and 1-year PFS was reached in 92% of cases. Previously treated patients with *RET* fusions presented 79% of ORR, 5% of complete responses, median PFS of 20.1 months, and 1-year PFS in 64% of cases [85]. The *RET* fusion-positive NSCLC patients presented the following results: in the platinum-based previously treated group the ORR was 64%, CR was observed in 2% of patients, mPFS was 16.5 months, and 1-year PFS was reached in 66% of patients; in the treatment-naïve group the ORR was 85%, mPFS was not reached, but the 1-year PFS was 75% [90].

The most common grade 3/4 treatment-related adverse events in MTC patients from the LIBRETTO-001 study were as follows: hypertension (12%), increased ALT (11%), increased AST (8%), diarrhea (3%), and prolonged QT (2%). The NSCLC *RET* fusion-positive patients presented similar, however, less common and more various, grade 3/4 treatment-related AEs than MTC patients: hypertension (9%), increased ALT (9%), increased AST (6%), prolonged QT (2%), diarrhea, rash, vomiting, and pyrexia (1% respectively). Only 2% of patients discontinued the treatment due to AEs [85,90].

The results and adverse events of both LIBRETTO 001 and ARROW studies are summarized in Table 1.

There are two ongoing LIBRETTO studies with selpercatinib: LIBRETTO-531 comparing selpercatinib and vandetanib/cabozantinib in MTC patients, and LIBRETTO-431 comparing selpercatinib and chemotherapy in NSCLC patients [91,92].

Although both drugs have been studied in adult thyroid and lung cancer patients, there are very few reports regarding the effectiveness of RET selective inhibitors in children. The first study on five young patients with *RET*-altered different tumors showed very promising results, with partial response or stable disease after selpercatinib treatment [93]. A study on six MEN2 pediatric patients treated with selpercatinib also presented very good results, with little toxicity [94]. There are two ongoing clinical trials regarding selpercatinib in pediatric patients with different tumors, and central nervous system metastases (ClinicalTrials.gov identifiers: NCT03899792 (LIBRETTO-121) and NCT04320888).

Studies showed that both of the approved selective RET inhibitors present effectiveness in CNS metastases treatment. In the ARROW trial, pralsetinib showed 56% of intracranial metastases responses and 3 patients with CR [82]. The analysis of the LIBRETTO-001 trial results showed that 82% NSCLC patients achieved intracranial response during the treatment with selpercatinib (23% complete responses and 59% partial responses). The intracranial disease control rate reached 100% [95]. Brain metastases of thyroid carcinomas are rare, but they can be also effectively treated with RET inhibitors. Selpercatinib showed effectiveness in a patient with *RET* M918T mutation, with multiple brain lesions, previously treated with vandetanib. After 8 weeks of treatment, the patient presented an almost complete response in brain metastases, which lasted 36 weeks until a new lesion occurred [96]. Tsui et al. presented a case of a patient treated with pralsetinib, which showed an extracranial response, but developed leptomeningeal progression, which was subsequently successfully treated with selpercatinib [97]. Such a case shows that *RET*-altered patients can be effectively treated interchangeably with selective RET inhibitors, depending on the type of lesion and its localization, although there is a need for large-scale studies.

## 9. Next Generation Selective RET Inhibitors

There are a few new RET-specific inhibitors under clinical trials. TPX-0046, which is a macrocyclic, VEGFR2/KDR sparing, RET/SRC inhibitor, showed promising results in studies on *RET*-altered cell lines. It presented very high activity in *KIF5B-RET* fusion cells carrying solvent front mutations G810C/R/S, which are the cause of resistance to pralsetinib and selpercatinib treatment [98]. A Phase1/2 clinical trial is underway (ClinicalTrials.gov identifier: NCT04161391). Another novel molecule BOS172738 showed high potency in both wild-type and gatekeeper mutant tumors. The result of the study presented an ORR of 33% in the NSCLC group and 44% in the MTC group, with one observed CR [99]. TAS0953/HM06 also presented effectiveness in pre-clinical analyses, showing cancer cells growth inhibition in cell lines with *RET* solvent front mutations [100]. LOXO Oncology is currently studying the second generation of selective RET inhibitors, with the ability to overcome both gatekeeper and solvent front mutations of *RET* gene [101].

The pathways activated by RET and other receptor kinases, including currently used and studied inhibitors, their targets, and possible reasons of resistance, are shown in Figure 2.

## 10. Resistance to RET Selective Inhibitors

It is a common observation, that patients undergoing molecularly targeted therapies progress in a mechanism of treatment resistance. This effect is usually due to occurrence of secondary mutations, which hinder the effectiveness of targeted drugs and cause relapse of the disease, however, other mechanisms can be involved. The V804X gatekeeper mutations in *RET* gene are the first cause of resistance to RET nonspecific MKIs [102,103]. Novel RET inhibitors are designed to bypass these mutations. The first resistance to selpercatinib was observed during the LIBRETTO-001 registration trial. The mechanism was based on the occurrence of secondary mutations in solvent front residues of *RET* gene, especially in G810 position, which caused substitution of glycine with large, charged or polar residues. This substitution resulted in altered protein structure and interference of selpercatinib binding to the ATP site. The observations were made in tumors of different histology, with different *RET* driver alterations. It was also observed in human *RET*-fusion xenografts that treatment with selpercatinib may induce the formation of gatekeeper mutation along with solvent front mutation [104]. Similar observations were made in the ARROW study concerning pralsetinib, where both C and N-lobe solvent front mutations were detected [105].

TPX-0046 was shown to be an effective drug in patients with both gatekeeper and solvent front mutations [106]. However, it is hypothesized that the presence of V804M gatekeeper mutation during the treatment with TPX-0046, might result in resistance to these second-generation RET inhibitors [107]. Laboratory analysis showed that, apart from the G810X C-lobe solvent front mutation, the N-lobe L710V/I mutation might cause resistance to pralsetinib but not selpercatinib. In animal models presenting the *KIF5B-RET*(L710V/I) changes, pralsetinib was shown to be blocked by the mutation creating a stronger steric clash and inhibiting the protein-drug interaction [108]. The possibility of resistance occurrence to first-generation RET-specific inhibitors is also presented in the case of mutations in residues Y806 (the hinge region) and V738 (the β2 strand of RET), which are involved in the binding of these drugs. The study has shown that Y806C/N and V738A mutations caused resistance to both selpercatinib and pralsetinib [89]. There have also been reports of a new *KHDRBS1*-*NTRK3* (K8; N1) fusion occurring as a resistance mechanism, in a *KIF5B-RET* positive lung cancer patient after treatment with selpercatinib [109].

The other mechanism of kinase inhibitors resistance is the clonal selection of tumor cells with alternative activating alterations emerging during the treatment process, most commonly MAPK signaling pathway activating mutations or amplifications, mainly *EGFR*, *KRAS*, *NRAS* or *BRAF* mutations, or *MET* and *FGFR1* amplification [56,110,111]. MET amplification was shown to be one of the reasons for the ineffectiveness of selpercatinib treatment (PFS of a maximum of 6 months). Combining treatment with crizotinib (ALK/ROS1/MET inhibitor) and selpercatinib allowed to overcome the resistance, even though for a short time period due to adverse effects [112].

## 11. Conclusions

RET alterations occur quite frequently as driving mutations or rearrangements in thyroid tumors, but also non-small cell lung cancer and other solid tumors. Abnormal RET proteins represent a target for multiple kinase inhibitors, but their specificity is far from satisfactory. This has recently become a point of access for new targeted therapies such as selective RET inhibitors, e.g., selpercatinib and pralsetinib, which have proven to be safe and effective in both MTC and NSCLC patients. However, issues have already been observed which need to be addressed. First of all, there is the occurrence of secondary RET alterations which cause resistance to first-generation RET inhibitors, creating a niche for development of second-generation drugs. Second, there is a need for large scale studies on combination therapies, including selpercatinib or pralsetinib and other therapeutics targeted to secondary mutations/fusions or amplifications occurring as a result of single-drug treatment. The ongoing clinical trials of second-generation RET inhibitors are promising, but we are still waiting for the results.

## Figures and Tables

**Figure 1 cancers-14-05298-f001:**
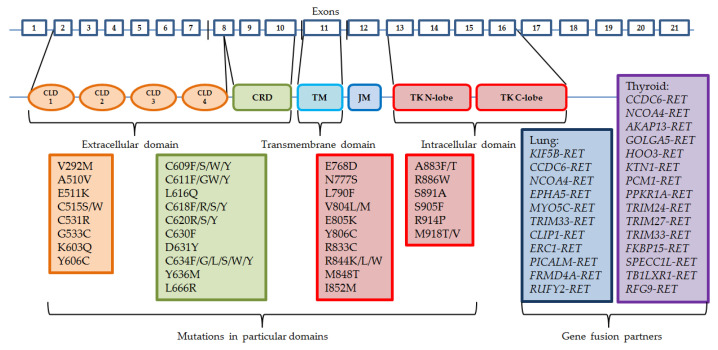
Structure of *RET* gene and the most common mutations and rearrangements. CLD—cadhedrin-like domain, CRD—cystein-rich domain, TM—trans-membrane domain, JM—juxta-membrane domain, TK—tyrosine kinase domain.

**Figure 2 cancers-14-05298-f002:**
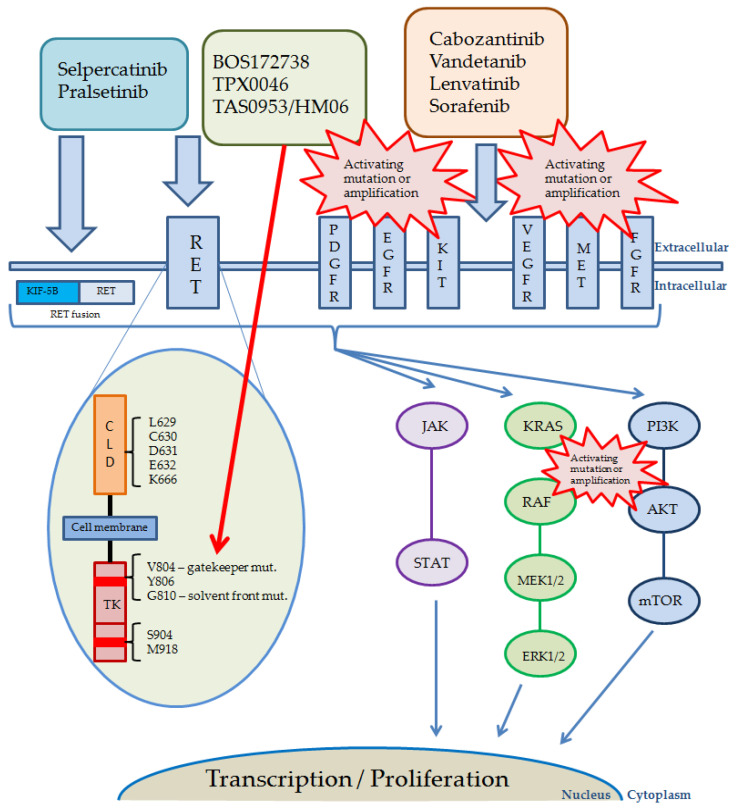
Pathways activated by RET and other receptor kinases, including drugs currently used. Red bursts present possible reasons and localization of the most common alterations causing RET-inhibitors resistance. The red arrow shows targets of novel, next generation RET inhibitors. Abbreviations: AKT, protein kinase B; KIT, KIT proto-oncogene receptor tyrosine kinase; MET, mesenchymal epithelial transition factor; CLD, cadherin-like domains; EGFR, epidermal growth factor receptor; ERK, extracellular signal-regulated kinases; FGFR, fibroblast growth factor receptor; JAK, Janus kinase; KIF-5B, anti-kinesin family member 5B; KRAS, Kirsten rat sarcoma 2 viral oncogene homolog; MAPK (MEK), mitogen-activated protein kinase kinase; ERK, extracellular signal-regulated kinase; mTOR, mammalian target of rapamycin; PDGFR, platelet-derived growth factor receptor; PI3K, phosphatidylinositol 3-kinase; RAF, rapidly accelerated fibrosarcoma; RET, rearranged during transfection; STAT, signal transducer and activator of transcription; TK, tyrosine kinase; VEGFR, vascular endothelial growth factor receptor.

**Table 1 cancers-14-05298-t001:** Registrations, effectiveness and adverse events of novel *RET*-specific inhibitors in NSCLC and thyroid cancers patients. NSCLC—non-small cell lung cancer, MTC—medullary thyroid carcinoma, ORR—overall response rate, DOR—duration of response, PFS—progression free survival, CR—complete response, NR—not reached, NE—not evaluated, ALT—alanine aminotransferase, AST—aspartate aminotransferase, AE—adverse events.

Drug		Selpercatinib (LOXO-292)	Pralsetinib (BLU-667)
Trial		LIBRETTO-001/Multicenter, Phase 1/2	ARROW/Multicenter, Phase 1/2
**Date of FDA approval/type of approval**		8 May 2020 for Adult patients with metastatic, *RET* fusion-positive NSCLC Adult and pediatric (≥12 years of age) patients with advanced or metastatic, *RET*-mutant MTC who require systemic therapy advanced or metastatic *RET* fusion-positive thyroid cancer who require systemic therapy and who are radioactive iodine refractory [76]	4 September 2020 for Adult patients with metastatic *RET* fusion-positive NSCLC 1 December 2020 for Adult and pediatric (≥12 years of age) patients with advanced or metastatic RET-mutant MTC who require systemic therapy RET fusion-positive thyroid cancer who require systemic therapy and who are radioactive iodine refractory [77]
**Clinical trial results**	NSCLC	***RET*-positive NSCLC patients** [90] Previously treated with platinum-based chemotherapy (*n* = 105) ORR—64%, CR—2% (*n* = 2), mDOR—17.5 months, mPFS—16.5 months, 1-year PFS—66% Previously untreated (*n* = 39) ORR—85%, CR—0%, mDOR—NE, mPFS—NE, 1-year—PFS 75%	***RET*-positive NSCLC patients** [82] Previously treated with platinum based chemotherapy (*n* = 87) ORR—61%, CR—6%, mDOR—NR, mPFS—17.1 months Previously untreated (*n* = 27) ORR—70%, CR—11%, mDOR—9 months, mPFS—9.1 months
	Thyroid cancer	***RET*-positive thyroid cancer patients** [85] *RET*-mutant MTC, previously treated (*n* = 55) ORR—69%, CR—9%, mDOR—NE, mPFS—NE, 1-year PFS—82% *RET*-mutant MTC, previously untreated (*n* = 88) ORR—73%, CR—11%, mDOR—22 months, mPFS—23.6 months, 1-year PFS—92% *RET* fusion-positive thyroid cancer, previously treated (*n* = 19) ORR—79%, CR—5%, mDOR—18.4, mPFS—20.1 months, 1-year PFS—64%	***RET*-positive thyroid cancer patients** [84] *RET*-mutant MTC, previously treated (*n* = 55) ORR—60%, CR—2%, mDOR—NR, mPFS—NR Treatment naïve, *RET*-mutant MTC (*n* = 21) ORR—71%, CR—5%, mDOR—NR, mPFS—NR *RET* fusion-positive thyroid cancer, previously treated (*n* = 9) ORR—89%, CR—0%, mDOR—NR, mPFS—NR
**Adverse events**		**NSCLC arm** Grade 1 & 2—39%, grade 3 & 4—58% **Thyroid cancer arm** Grade 1& 2—32%, grade 3 & 4—66%	**NSCLC arm** All toxicities—93%, grade 3 & 4—48% **Thyroid cancer arm** Grade 1 & 2—43%, grade 3 & 4—53%
**Most common grade 3 & 4 AEs related to treatment**	NSCLC	Hypertension (9%), increased ALT (9%), increased AST (6%), diarrhea (2%), prolonged QT (2%), constipation (1%), rash (1%), vomiting (1%), pyrexia (1%), thrombocytopenia (1%)	Neutropenia (19%), hypertension (11%), anemia (10%), decreased white blood cell count (6%), lymphopenia (5%), increased ALT (3%), increased AST (3%), phosphokinase (3%). thrombocytopenia (3%), Elevated blood creatine phosphokinase (3%), pneumonia (<4%)
	Thyroid cancer	Hypertension (12%), increased ALT (11%), increased AST (8%), diarrhea (3%), prolonged QT (2%), fatigue (1%), headache (1%), weight increased (1%)	Hypertension (17%), neutropenia (14%), lymphopenia (12%), anemia (10%), decreased white blood cell count (8%), increased blood creatine phosphokinase (5%), asthenia (4%), pneumonitis (3%), diarrhoea (2%)

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
