# Peer review of "RET Proto-Oncogene—Not Such an Obvious Starting Point in Cancer Therapy"

_cancers, 2022, doi:10.3390/cancers14215298_

Round 1

Reviewer 1 Report

In the present review, Kucharczyk et al. explore the hot field of RET targeted therapies in thyroid and lung cancers. To this purpose, they describe the current drugs approved FDA and EMA to fight RET activity in thyroid and lung tumors underscoring the low effectiveness of these therapeutics and the need of novel and more appropriate drugs to obtain more impactful targeted molecules with long-lasting response. Selpercatinib and pralsetinib are reported by the authors as more recent drugs with tolerable side effects and good effectiveness, even if with low durable response due to the emergence of novel RET mutations or tyrosine kinase receptor alternative pathways. Finally, the authors report the last and more recent RET inhibitors developed to overcome selpercatinib and pralsetinib resistance and the ongoing clinical trials.

The review not only is well presented and organized, but the authors also give an exhaustive and updated overview of RET targeted therapy state of art. Nevertheless, it should be further improved to be suitable for publication on Cancers.

 Major points:

1.      The authors include only one figure and one table in the review: too reductive. One of the most appreciated feature of a review is its promptness to give clear messages to the reader that is trying to figure out the RET inhibitors world. Thus, a figure depicting the most used inhibitors and the corresponding targeted mutations should be added.

    Moreover, it should be added a figure showing the different targets of RET signal transduction affected by RET inhibitors and the alternative pathways that activate these targets overcoming RET inhibition in cancer.

    A conclusive figure depicting the novel drugs action and how they can hopefully avoid alternative pathways activation.

2.      It is not clear to the referee why the paragraphs “Detection of alterations” are repeated four times in the review, three times in succession and one time separated by other paragraphs: this issue induces confusion to the reader. The authors should re-organize these paragraphs.

3.      “MKIs approved by the FDA and EMA for the treatment of radioiodine refractory differentiated thyroid cancer (DTC) are sorafenib and lenvatinib [68,69], however, lenvatinib has been proven to be more effective than sorafenib (mPFS reached 13,3 vs. 35,3 months, respectively) [70]”.

I suppose it would be written “mPFS reached 35,3 vs. 13,3 months, respectively”.

Author Response

As Authors, we are grateful for the valuable review of our article. We hope that the comments and remarks will contribute to improving the quality of the manuscript. At the same time, we would like to thank the reviewer for the time taken to complete the review. Responses to the reviewers comments can be found below, highlighted in blue. All changes to the manuscript are visible in the uploaded reviewed version of the manuscript.

Reviewer 1.

In the present review, Kucharczyk et al. explore the hot field of RET targeted therapies in thyroid and lung cancers. To this purpose, they describe the current drugs approved FDA and EMA to fight RET activity in thyroid and lung tumors underscoring the low effectiveness of these therapeutics and the need of novel and more appropriate drugs to obtain more impactful targeted molecules with long-lasting response. Selpercatinib and pralsetinib are reported by the authors as more recent drugs with tolerable side effects and good effectiveness, even if with low durable response due to the emergence of novel RET mutations or tyrosine kinase receptor alternative pathways. Finally, the authors report the last and more recent RET inhibitors developed to overcome selpercatinib and pralsetinib resistance and the ongoing clinical trials.

The review not only is well presented and organized, but the authors also give an exhaustive and updated overview of RET targeted therapy state of art. Nevertheless, it should be further improved to be suitable for publication on Cancers.

 Major points:

  1. The authors include only one figure and one table in the review: too reductive. One of the most appreciated feature of a review is its promptness to give clear messages to the reader that is trying to figure out the RET inhibitors world. Thus, a figure depicting the most used inhibitors and the corresponding targeted mutations should be added.

    Moreover, it should be added a figure showing the different targets of RET signal transduction affected by RET inhibitors and the alternative pathways that activate these targets overcoming RET inhibition in cancer.

    A conclusive figure depicting the novel drugs action and how they can hopefully avoid alternative pathways activation.

We have included ‘Figure 2’, which includes pathways activated by RET and other receptor kinases, drugs that are currently used and those studied, and the possible genetic changes that might be the reason for the resistance to these drugs.

The text involved in the manuscript, under paragraph 9:

‘The pathways activated by RET and other receptor kinases, including currently used and studied inhibitors, their targets and possible reasons of resistance, is shown on Figure 2.’

‘Figure 2. Pathways activated by RET and other receptor kinases, including drugs currently used. Red bursts present possible reasons and localization of the most common alterations causing RET-inhibitors resistance. The red arrow shows targets of novel, next generation RET inhibitors. Abbreviations: AKT, protein kinase B; KIT, KIT proto-oncogene receptor tyrosine kinase; MET, mesenchymal epithelial transition factor; CLD, cadherin-like domains; EGFR, epidermal growth factor receptor; ERK, extracellular signal-regulated kinases; FGFR, fibroblast growth factor receptor; JAK, Janus kinase; KIF-5B, anti-kinesin family member 5B; KRAS, Kirsten rat sarcoma 2 viral oncogene homolog; MAPK (MEK), mitogen-activated protein kinase kinase; ERK, extracellular signal-regulated kinase; mTOR, mammalian target of rapamycin; PDGFR, platelet-derived growth factor receptor; PI3K, phosphatidylinositol 3-kinase; RAF, rapidly accelerated fibrosarcoma; RET, rearranged during transfection; STAT, signal transducer and activator of transcription; TK, tyrosine kinase; VEGFR, vascular endothelial growth factor receptor‘

  1. It is not clear to the referee why the paragraphs “Detection of alterations” are repeated four times in the review, three times in succession and one time separated by other paragraphs: this issue induces confusion to the reader. The authors should re-organize these paragraphs.

This is clearly our oversight during the preparation of the final manuscript. The names of paragraphs 6, 7 and 10 have been changed according to their content:

  1. Immunotherapy in tumors with genetic changes of RET gene
  2. Treatment of RET-altered cancers with multiple kinase inhibitors
  3. Resistance to RET selective inhibitors

  1. “MKIs approved by the FDA and EMA for the treatment of radioiodine refractory differentiated thyroid cancer (DTC) are sorafenib and lenvatinib [68,69], however, lenvatinib has been proven to be more effective than sorafenib (mPFS reached 13,3 vs. 35,3 months, respectively) [70]”.

I suppose it would be written “mPFS reached 35,3 vs. 13,3 months, respectively”.

The order of the data has been changed accordingly.

Reviewer 2 Report

Recently I was invited to review an interesting paper entitled: RET proto-oncogene – not such an obvious starting point in cancer therapy. The paper is well written and easy to understand. I think it is important to discuss molecular aberrations in non-small cell lung cancer, especially in the context of novel therapies leading to new treatment options.
In the introduction authors sufficiently describe the background of the molecular targets in NSCLC with special attention given to tyrosine kinases.
In the following parts of the manuscript, the authors discuss RET protein and its activity. The part about the somatic RET gene rearrangements is supplemented by an informative Figure 1. The part of the manuscript describing the detection of the RET alterations is clearly due to the preceding chapter describing the mutations.
I appreciate the part of the manuscript related to drug resistance. It is an important issue and is adequately described by the authors.
The references are up-to-date and well chosen.
Minor remarks.

Chapters 6, 7, and 10 are labeled with a clear mistake. Please correct.

Author Response

As Authors, we are grateful for the valuable review of our article. We hope that the comments and remarks will contribute to improving the quality of the manuscript. At the same time, we would like to thank the reviewer for the time taken to complete the review. Responses to the reviewers comments can be found below, highlighted in blue. All changes to the manuscript are visible in the uploaded reviewed version of the manuscript.

Reviewer 2.

Recently I was invited to review an interesting paper entitled: RET proto-oncogene – not such an obvious starting point in cancer therapy. The paper is well written and easy to understand. I think it is important to discuss molecular aberrations in non-small cell lung cancer, especially in the context of novel therapies leading to new treatment options.
In the introduction authors sufficiently describe the background of the molecular targets in NSCLC with special attention given to tyrosine kinases.
In the following parts of the manuscript, the authors discuss RET protein and its activity. The part about the somatic RET gene rearrangements is supplemented by an informative Figure 1. The part of the manuscript describing the detection of the RET alterations is clearly due to the preceding chapter describing the mutations.
I appreciate the part of the manuscript related to drug resistance. It is an important issue and is adequately described by the authors.
The references are up-to-date and well chosen.
Minor remarks.

Chapters 6, 7, and 10 are labeled with a clear mistake. Please correct.

This is clearly our oversight during the preparation of the final manuscript. The names of paragraphs 6, 7 and 10 have been changed according to their content:

  1. Immunotherapy in tumors with genetic changes of RET gene
  2. Treatment of RET-altered cancers with multiple kinase inhibitors
  3. Resistance to RET selective inhibitors.

Reviewer 3 Report

Mutations and fusion involving RET, a transmembrane tyrosine receptor kinase is found in multiple cancers. However, Multiple kinase inhibitors have shown little or no efficiency in diseases with RET gene alteration. In this review authors have overviewed the effectiveness of new RET-specific inhibitors e.g., Selpercatinib and pralsetinib in clinical trials involving NSCLC and thyroid cancer patients. Authors have also discussed the molecular function of RET receptor and gene alterations commonly associated with RET gene in various cancers. They have briefly touched on the methods to detect RET alterations and effectiveness of immunotherapies in RET altered malignancies.  They have also discussed emergence of resistant mutations and activation of different signaling pathways as possible mechanism for drug resistance in RET altered cancers.  Overall review is concise and neatly written and I would recommend it for publications with minor changes. 

Author Response

As Authors, we are grateful for the valuable review of our article. We hope that the comments and remarks will contribute to improving the quality of the manuscript. At the same time, we would like to thank the reviewer for the time taken to complete the review. Responses to the reviewers comments can be found below, highlighted in blue. All changes to the manuscript are visible in the uploaded reviewed version of the manuscript.

Reviewer 3.

Mutations and fusion involving RET, a transmembrane tyrosine receptor kinase is found in multiple cancers. However, Multiple kinase inhibitors have shown little or no efficiency in diseases with RET gene alteration. In this review authors have overviewed the effectiveness of new RET-specific inhibitors e.g., Selpercatinib and pralsetinib in clinical trials involving NSCLC and thyroid cancer patients. Authors have also discussed the molecular function of RET receptor and gene alterations commonly associated with RET gene in various cancers. They have briefly touched on the methods to detect RET alterations and effectiveness of immunotherapies in RET altered malignancies.  They have also discussed emergence of resistant mutations and activation of different signaling pathways as possible mechanism for drug resistance in RET altered cancers.  Overall review is concise and neatly written and I would recommend it for publications with minor changes. 

Round 2

Reviewer 1 Report

The manuscript is now suitable for publication on Cancers.